# Variability in Language and Literacy Outcomes Among Deaf Elementary Students in a National Sample

**DOI:** 10.3390/bs15081100

**Published:** 2025-08-13

**Authors:** Kimberly Wolbers, Hannah Dostal, Kelsey Spurgin

**Affiliations:** 1Theory and Practice in Teacher Education, University of Tennessee, Knoxville, TN 37996, USA; 2Department of Curriculum and Instruction, University of Connecticut, Storrs, CT 06269, USA; hannah.dostal@uconn.edu; 3Department of Special Education, Ball State University, Muncie, IN 47306, USA; kelsey.spurgin@bsu.edu

**Keywords:** deaf, elementary, reading, writing, literacy

## Abstract

This study examined the literacy outcomes of 368 deaf elementary students in the United States, focusing on reading and writing performance and their connections with demographic and language variables. Standardized assessment data were analyzed from students in grades 3–6. Results indicated wide variability in reading and writing performance, from scores at a standard deviation above the mean to more than 3 below the mean. There were demonstrated disparities in mean literacy outcomes based on disability status. A strong positive correlation was found between reading and writing scores, suggesting interconnected development of literacy skills. Notably, writing outcomes were consistently higher than reading across analyses. Performance trajectories differed by grade, with literacy gaps widening over time. Gender, race, and hearing level explained 2–3% of the variance in literacy outcomes, while language proficiency (in ASL and/or spoken English) and phonological knowledge (fingerspelled and/or spoken) predicated 55–63% of the models. These findings highlight the need for early accessible language exposure and responsive literacy instruction aligned with deaf learners’ language strengths.

## 1. Literature Review

Deaf and hard-of-hearing children (hereafter referred to as deaf) represent a heterogeneous population with varied linguistic and cognitive backgrounds. This variability reflects complex traits shaped by individual differences and their environments ([62]). Research on deaf children’s development must consider factors beyond hearing level. Cultural identities ([8]), multilingual language environments ([14]), and the presence of additional disabilities ([7]) all play a pivotal role in shaping a deaf child’s language and cognition. This diversity leads to variability in outcomes like language expression, social interaction, cognition, and executive functioning, which have major implications for education. This is especially true as inequities in academic attainment persist ([25]) despite advancements in crucial supports like newborn hearing screening and identification ([75]) and early intervention ([47]). Research that centers these diverse experiences works to expand developmental theory and improve instructional approaches for a broad range of learners.

### 1.1. Language Foundations

Early exposure to language-rich environments is essential for literacy development. For deaf children, consistent access to signed or spoken language is vital for growth in literacy and cognition. There are multiple pathways to language proficiency for deaf children ([31]). Some recommend against learning sign language alongside spoken language ([28]) due to misconceptions about neural plasticity ([13]), language acquisition ([54]), and hearing parents’ capacity to learn and model sign language ([41]). However, research shows that both signed and spoken input support strong language foundations and cognitive growth ([15]). Even caregivers who are learning sign language alongside their deaf child can offer ample language input to support age-appropriate vocabulary development ([10]; [21]), reinforcing the importance of early, accessible communication, regardless of modality ([2]).

Along with newborn hearing screening and early intervention services, hearing technology has seen a dramatic increase in sophistication and availability, leading to increasingly earlier access to auditory input ([51]). It is important, however, to distinguish between quality access to technology and quality access to actual language, as the mere use of a device does not guarantee optimal language input ([9]), nor should early access to technology be considered the dominant variable influencing language growth ([19]).

Early and accessible language fosters cognitive–linguistic skills that enhance later learning. For example, consistent language input in a deaf child’s early years is crucial for developing receptive vocabulary—the building blocks of language and literacy ([2]). Additionally, early communication with caregivers develops visual attention, essential for visual language acquisition ([20]; [60]) and executive function skills ([33]). Research has shown how deeply interconnected these processes are, pointing to language as a key mediator in developing executive functioning ([6]). Early interactions also foster theory of mind ([48]; [58]) and working memory ([45]), both linked to early literacy ([65]). Foundational language skills shape later reading development, including vocabulary knowledge ([10]) and orthographic awareness ([1]), underscoring the importance of early and accessible language for cognition and literacy outcomes ([24]).

### 1.2. Literacy Foundations and Performance

Given the diverse backgrounds and language experiences of deaf children, literacy performance data should be collected and interpreted through a context-sensitive lens ([23]; [29]). Recent large-scale research points to the expressive language diversity of deaf children to explain diverse developmental paths in literacy. Deaf students show similarities in the underlying skills needed for early reading, such as a strong expressive/receptive language foundation and sublexical processing of word reading; however, these skills vary in application by visual or spoken language modalities ([40]).

When provided a linguistically accessible reading curriculum, deaf children demonstrate progress in early reading skills. [3] ([3]) collected language and literacy data with 336 deaf children in kindergarten through second grade at the beginning and end of one academic year. The sample was organized into three language modality groups: spoken only, sign only, and bimodal. Those with functional hearing showed growth in English and spoken phonological knowledge. Signing deaf students demonstrated growth in American Sign Language (ASL) syntax and fingerspelling phonological knowledge. Both types of phonological knowledge are pathways to decoding words using sublexical processes. When phonological knowledge (whether spoken or fingerspelled) aligns with the way words are stored in students’ cognitive systems (as sound-based or visual-based), comprehension is bolstered ([40]). Considering the extensive heterogeneity of this population, these findings highlight the complex ways that individual experiences (e.g., early language access, cognitive development, cultural background) and language dispositions interact with learning environments to shape language and literacy outcomes.

The last decade of research on language and literacy outcomes brings a growing awareness of the varied developmental trajectories of deaf children. It is becoming increasingly clear that the variability in performance does not reflect a deficit model (i.e., the idea that deaf children plateau at a certain point in their capability to develop new reading skills). Instead, multiple valid developmental pathways are present, which challenge the belief that there is a single correct normative trajectory ([12]). [12] ([12]) analyzed NWEA MAP reading and math assessment outcomes for 351 deaf students attending schools for the deaf, as well as a group of hearing students, over a five-year period. Second- through eighth-grade deaf students demonstrated steady growth, making similar annual gains in all domains as their hearing peers. This asset-oriented perspective invites researchers and educators to dig deeper into patterns of consistent growth and consider diverse strengths in the ways that deaf children learn.

The development of writing skills is also deeply intertwined with early language acquisition. Writing is a demanding and complex process that draws upon numerous linguistic and cognitive skills, including vocabulary, syntax, idea organization, and metalinguistic awareness. Unlike reading, in which language is received through decoding, writing requires students to draw from their language repertoire to encode novel ideas. This means that a child’s writing experiences are directly mediated by their unique idiolects. Just as emergent readers benefit from instructional strategies that align with their sound-based and/or sign-based cognitive systems ([40]), early writers benefit from explicit instruction with translating ideas and encoding language by drawing upon and expanding their current linguistic structures ([71]). For monolingual children developing listening and spoken language, teachers support writing skills by attending to students’ expressive and receptive language development, along with explicit instruction in how they represent that language in written form ([30]). Bilingual and multilingual students naturally draw from all of their language resources during the composing process. When a student communicates through multiple languages and modalities (e.g., signed, written, spoken), translanguaging theory explains that they do not compartmentalize their languages ([27]); rather, they are “working from their single, unified linguistic system during meaning-making processes, tapping into all words, linguistic features, and semiotic resources they know to express ideas in print” ([35]). When learning a new target language, multilingual students benefit from writing instruction that validates and engages all of their linguistic resources while heightening their metalinguistic awareness.

Through specialized language instruction, students are empowered to expand and flexibly draw on a broader range of language practices to communicate effectively with various audiences and for diverse purposes. [63] ([63]), for instance, studied the syntactic development of written Chinese (a non-alphabetic language) among deaf elementary students while providing “flexible code choice” (p. 693) in Hong Kong Sign Language and spoken Cantonese instruction within a sign bilingual co-enrollment (SLCO) setting. In this study, hearing (multilingual) students enrolled in the same SLCO classrooms served as the control group. The researchers found that, despite modality and structural differences across Hong Kong Sign Language, spoken Cantonese, and written Chinese, the deaf students developed written Chinese grammatical skills in a similar order to their hearing peers. While these syntactic skills were initially gained at a slower rate compared to their hearing peers, the deaf students narrowed the gap over time, showing consistent progress across elementary grades.

Another concern discussed in the field is whether the acquisition of a signed language can interfere with the development of written language. Elements from one language sometimes appear in another, such as phonological or grammatical features of Spanish or ASL within English text ([35]; [72]). These applications are an expected and natural feature of multilingual development ([57]) and provide evidence of a writer accessing their whole linguistic repertoire to compose and share their ideas ([26]). Like a web, a writer’s linguistic repertoire is comprised of many linguistic resources that span across modalities and languages, and it grows organically through meaningful interactions with others. Writing for different authentic purposes and audiences further encourages the curation and adaptation of language features that support a writer’s goals. In a qualitative case study that centered on the writing of three young deaf siblings across 10 years, [35] ([35]) examined the writing development and translanguaging features within 28 written samples. These writers, aged three-to-ten years old, were raised and educated in ASL/written English environments, and their writing demonstrated all of the stages of emergent writing development. Translanguaging features (e.g., embedded applications of linguistic structures) were identified in earlier writings, demonstrating how the young writers leveraged their linguistic repertoires as they developed new language skills. As they grew older and their metalinguistic awareness matured, these features decreased as they deployed linguistic resources that were increasingly appropriate to their intended audience, purpose, format, and language.

Given this perspective of how deaf students use their linguistic resources to construct meaning, there are important implications for how teachers and researchers approach the evaluation of their writing. [27] ([27]) propose that purposeful assessments should be designed through the lens of translanguaging theory; however, many of the tools and approaches available to teachers are normed and designed for hearing, monolingual learners ([37]). This gap between theory and practice points to the significance in conducting broad-ranging research that captures how deaf students develop language skills and communicate through their writing. In an analysis of a large collection of writing samples using a functional grammar framework ([38]), the authors map the progression of syntactic structures used by students, providing a view into their writing development. They found that emergent writers tended to express ideas through one-word nouns and simple sentences, while more fluent writers expanded noun phrases, used adverbs, and embedded clauses. While writing outcomes for deaf students demonstrate wide variability, studying these patterns in development helps researchers and educators recognize the strengths and trajectories in all students’ writing. Meaningful and sustained progress is achievable when instruction and assessment tools are responsive to students’ linguistic experiences.

### 1.3. Demographic Factors and Literacy Outcomes

Research has examined how demographic factors such as hearing level, gender, race, and family hearing status relate to the literacy development of deaf students. Some studies have found that students with mild-to-moderate hearing levels ([64]) or unilateral hearing loss ([46]) perform better on standardized assessments than peers with severe hearing levels and bilateral hearing loss. Other studies, however, have shown that even a mild hearing difference can impact access to language and instruction in meaningful ways ([44]; [56]). Hearing technologies are utilized to enhance children’s access to spoken language, and the consistency of technology use has been identified as a key predictor of literacy skill development ([68]). Even so, researchers of the early language and literacy acquisition study found that among aided deaf and hard-of-hearing children who use spoken language, there are significant delays in early language development ([42]; [66]) and emergent literacy skills ([69]) compared to their hearing counterparts.

In terms of gender, research findings have been inconsistent or have mirrored broader trends in general education research ([49]; [74]). While disparities in educational access and outcomes persist along racial and socioeconomic lines ([50]), these factors are often entangled with broader systemic and contextual factors that go well beyond individual student traits. For example, in a study investigating vocabulary across 448 children who were deaf or hard of hearing and who had participated in early intervention, [76] ([76]) found that 41% of the variance in vocabulary performance was explained by a combination of adherence to EHDI guidelines, being younger in age, having no additional disabilities, having a mild-to-moderate hearing level, higher maternal education, and having deaf or hard-of-hearing parent(s). Many other studies have demonstrated favorable literacy outcomes for deaf children who have signing deaf parents ([59]). These outcomes have been attributed to early and consistent access to a natural signed language rather than to parental hearing status alone ([21]).

Studying large datasets provides a valuable lens for understanding the diversity within the deaf student population. Smaller or more homogenous samples are able to control for or focus on certain variables, but a large sample more broadly demonstrates language and literacy outcomes across various educational contexts and diverse learner backgrounds. With large-scale data, researchers can examine how learner variables interact with standardized measures of reading and writing and related subskills. In this study, we analyze results from standardized literacy assessments to explore these interactions. Rather than simply comparing deaf students to normative expectations, our goal is to better understand and examine the variability within this population.

In the current study, we investigated the following research questions:Research Question 1: What are the literacy scores of deaf elementary students in the US? What is the relationship between writing and reading outcomes?Research Question 2: How do literacy scores vary across student demographic groups?Research Question 3: How do students’ literacy scores vary based on language proficiency levels?Research Question 4: To what extent do language proficiency and phonological knowledge (spoken or fingerspelled) explain variance in students’ literacy scores?

## 2. Method

The purpose of this study was to investigate the language and literacy outcomes of a sample of deaf elementary students in the United States, with particular attention to how demographic, linguistic, and educational factors interact with academic performance. Building on research that highlights the variability in language acquisition and academic development among deaf children, this study aimed to identify patterns in literacy skills by using standardized assessments commonly used in educational and research contexts. This allowed us to explore how students’ language experiences may be related to their literacy outcomes.

The data used in this study were collected as part of a larger project funded by the Institute of Education Sciences (IES), which focused on Strategic and Interactive Writing Instruction (SIWI; [70]), an approach to writing instruction for deaf students. All assessments were administered at the beginning of the academic year, prior to the implementation of SIWI, and thus were not influenced by participation in the SIWI program. A key delimitation of this study is that it includes only deaf students who received writing instruction from a deaf education teacher for a minimum of 2 h per week. This was a necessary requirement for teacher participation in the larger study to ensure that all participating students had consistent access to the intervention, allowing us to explore the impact of SIWI on students’ literacy outcomes.

The study was conducted in accordance with the Declaration of Helsinki and approved by the Institutional Review Board of the University of Tennessee (UTK IRB-17-03771-XP, first approved 6 June 2017).

### 2.1. Participant Demographics

A total of 368 elementary students participated in the study, representing four grade levels. The distribution across grade levels and by gender, by racial and ethnic identity, and by disability are reported in Table 1. The mean age for third grade students was 8 years, 11 months (ranging from 7 years to 10 years, 9 months); fourth graders’ mean age was 9 years, 10 months (ranging from 8 years, 1 month to 10 years, 11 months); fifth graders’ mean age was 11 years (ranging from 9 years, 8 months to 13 years, 5 months); and sixth graders’ mean age was 11 years, 10 months (ranging from 9 years, 6 months to 14 years, 5 months).

#### 2.1.1. Educational Context

The 368 students in this study attended a range of educational settings that had various philosophies guiding language and communication. The majority (n = 220, 59.8%) attended deaf schools. An additional 128 students (34.8%) were in self-contained classrooms within public schools, while 20 students (5.4%) received itinerant services. Just over one-third of students (n = 127, 35%) were in ASL/English bilingual settings, all of which were located at schools for the deaf. Total communication was the most commonly reported approach, used with 140 students (38%). Overall, 70 students (19%) were in programs that used spoken language with sign support, and 31 students (8.4%) were in settings that used listening and spoken language only. Due to the inclusion criteria that students must be receiving at least 2 h of language and writing instruction from a deaf education teacher, the sample does not include students fully integrated into the general education setting. The United States (US) Department of Education’s [53] ([53]) estimates that among the broader deaf student population in the US, approximately 20.8% of students attend specialized schools or programs for the deaf, and over 75% of deaf students are educated in mainstream settings ([52]). The study sample is representative of those students who receive deaf education services in special and public school programs.

Data were available for 286 students regarding their prior school placement. Of these students, 146 students (51%) had only ever attended their current placement, while 140 students (49%) had been previously enrolled in another school or program different by setting and/or philosophy. Data were not available for 82 students. Regarding length of enrollment at their current placement for the 286 students with available data, 34 (12%) were new transfers to their current placement, 31 (11%) had been enrolled for one academic year, another 31 (11%) for two years, and 30 students (10%) for three years. In total, 126 students (44%) had attended their current school for three years or less, highlighting a high degree of student mobility within the sample. Of the students who had previously changed schools/programs (n = 140), they are now being served in deaf schools (n = 93), self-contained classes in public schools (n = 43), and itinerant services (n = 4). For those now served in deaf schools, they had transferred from public schools (n = 57), home school, no school, or orphanages (n = 6), oral schools for the deaf (n = 4), signing schools for the deaf (n = 13), and unknown (n = 13). For those being served in self-contained classes in public schools, they had transferred from other public schools (n = 26), schools for the deaf (n = 2), and unknown (n = 15). For those being served in itinerant settings, they had transferred from other public schools (n = 3), while one student had both public school and school for the deaf experiences (n = 1). Based on these data, 71% of new arrivals to schools for the deaf come from public school programs, and approximately 9% of new arrivals to public schools come from schools for the deaf. 

Furthermore, attendance data indicated high absenteeism in the sample, with 107 students (29%) missing 10 or more school days in a school year and 47 of those students (7.8%) missing 20 or more days.

#### 2.1.2. Hearing Levels

The students’ hearing levels in dB (unaided and aided) and hearing technology (type and usage) were collected from the schools through a demographic survey. Hearing level was the pure tone average in the better ear reported on the students’ audiogram on file at the school. School personnel selected the category that reflected the students’ hearing level in dB—typical/slight (0–25 dB), mild (26–40 dB), moderate (41–55 dB), moderately severe (56–70 dB), severe (71–90 dB), and profound (91 dB or greater).

Among the 368 students in the study, hearing levels measured in the better ear without amplification ranged from typical/slight to profound hearing levels. Specifically, 8 students (2.2%) had typical to slight hearing levels, 15 students (4.1%) had mild hearing levels, 39 students (10.6%) had moderate hearing levels, 65 students (17.7%) had moderately severe hearing levels, 69 students (18.8%) had severe hearing levels, and 165 students (44.8%) had profound hearing levels. Data were missing for seven students.

Eighty-one students (22%) did not have amplification devices, while others used a range of devices. In total, 147 students (39.9%) used hearing aids, 39 (10.6%) had one cochlear implant, 35 (9.5%) used a cochlear implant in one ear and a hearing aid in the other, and 63 (17.1%) used bilateral cochlear implants. Amplification device data were missing for three students.

We report students’ best hearing levels based on either their amplified hearing (for 209 students who used hearing devices frequently or always, as reported by their teachers) or their unamplified hearing (for 89 students who did not use or have devices), with a total sample of 298 students. For 70 students (19.1%), this information was not reported. Overall, 54 students (14.7%) were reported to have typical to slight hearing loss (0–25 dB) and 57 (15.5%) mild hearing levels (26–40 dB). A total of 33 students (9%) had moderate hearing levels (41–55 dB), and 27 students (7.3%) were identified as having moderately severe hearing levels (56–70 dB). Finally, 30 students (8.2%) had severe hearing levels (71–90 dB), while 97 students (26.4%) were reported to have profound hearing levels (91 dB or greater).

#### 2.1.3. Language

Teachers assessed students’ language proficiency in both ASL and spoken English using a five-point Likert scale, where 1 indicated the ability to “express most anything” and 5 indicated the student “does not express anything in the language”. Teachers who assessed students’ language proficiency were fluent in the language(s) they evaluated. If the teacher of record was not fluent in one of the languages, a certified colleague who worked closely with the student provided the rating. In ASL, 100 students (27.2%) were described as able to express most things, while another 94 (25.5%) could express many things. A smaller group of 70 students (19%) was noted as having difficulty expressing many things, and 32 (8.7%) had difficulty expressing most things. Sixty-three students (17.1%) were reported as not using ASL at all, and proficiency information was missing for nine students. Spoken English proficiency followed a similar distribution, though slightly more students were reported as non-users. Sixty-four students (17.4%) were reported to be able to express most things in spoken English, and ninety-five (25.8%) could express many things. Others were described as having difficulty expressing many things (63 students, 17.1%) or most things (30 students, 8.2%), while 107 students (29.1%) did not use spoken English. Again, data were not reported for nine students. Taken together, 139 students (37.8%) were rated as able to express most anything in at least one of the two languages (see Table 2).

English was the most commonly reported home language, with over one-third of families (144 students, 39.1%) using it as the primary way they communicated with each other. ASL was used in 66 homes (17.9%), while 53 students (14.4%) used both English and ASL at home. Spanish was used in 33 students’ homes (9%) to communicate between family members and the child. There were 47 students (12.8%) whose language in the home was something other than English, ASL, and/or Spanish. Among the 15 additional languages identified, the most common was Bengali (4 students), followed by Portuguese and Sango (2 students each). Data about home language use were not reported for 25 students.

Out of the full student sample, 93 students (23.2%) reported having at least one deaf family member. Among these, 64 students indicated that they had one or two deaf caregivers, with some of these families also including deaf siblings. An additional 29 students reported having deaf siblings but not deaf caregivers. In contrast, the majority of students (247, 66.9%) did not have any deaf individuals in their home environment. Information on family hearing status was missing for 29 students (7.9%).

### 2.2. Data Collection

At the beginning of the academic year prior to instruction, students were given subtests that comprise the Broad Written Language and Broad Reading clusters of the Woodcock–Johnson IV Test of Achievement (WJ-IV) and the Peabody Picture Vocabulary Test-IV (PPVT). The assessments were administered by trained data collectors who were not the students’ classroom teachers, thereby reducing potential bias. All assessments were scored by the research team following the assessments’ standardized scoring protocols.

### 2.3. Measures

In this study, the WJ-IV Broad Reading and Broad Written Language Clusters, as well as the PPVT, were used for analyses.

#### 2.3.1. Woodcock–Johnson IV (WJ-IV)

The WJ-IV Broad Reading cluster includes the Letter–Word Identification subtest, Reading Fluency subtest, and Passage Comprehension subtest. These components evaluate a range of reading skills, from basic word recognition to comprehension of connected text. A cluster score serves as an indicator of overall reading performance. The WJ-IV Broad Written Language cluster includes the following subtests: Spelling, Sentence Writing, and Writing Samples. Together, these evaluate elements of students’ orthographic knowledge, syntactic development, and ability to generate written responses and provide a global indicator of writing performance.

Each WJ-IV subtest provides both a standard score and a W score. The standard score is norm-referenced, with a mean of 100 and a standard deviation of 15, and it cannot fall below a value of 40 due to the scoring structure of the assessment. In contrast, the W score represents a Rasch-derived equal-interval scale that does not have a truncated range, providing a more sensitive measure of student performance.

#### 2.3.2. Peabody Picture Vocabulary Test-IV (PPVT)

The PPTV test items were administered to students using fingerspelling, spoken English, or both, depending on the language(s) they used. Test administrators were instructed not to provide ASL signs or explanations. The fingerspelling test items were videos produced by a deaf adult with expertise in the use of fingerspelling in educational contexts. For spoken English administration, examiners said each word aloud while ensuring students had unobstructed visual access to their mouth. Both fingerspelling and spoken English were provided to bimodal students.

This administration approach broadened the utility of the test to serve as a proxy measure of students’ phonological knowledge (sound based or visual) and receptive vocabulary. Students needed to rely on their spoken phonological processing knowledge or fingerspelling phonological processing knowledge to retrieve words stored in their cognitive systems, either as spoken forms or as visually encoded representations through fingerspelling. The task then required linking these internally stored words to the corresponding pictures. Thus, performance on the PPVT reflected an integrated process involving phonological access (sound and/or visual) and vocabulary knowledge, highlighting the multimodal pathways through which deaf students may store and retrieve language.

Growth Scale Values (GSVs) are provided for the PPVT. The GSV provides an equal-interval scale score designed to track individual growth over time. GSVs are not norm-referenced and do not have ceiling or floor effects.

### 2.4. Data Analysis

To address research question 1, which focuses on describing the literacy scores of deaf elementary students and exploring the relationship between writing and reading outcomes, we first calculated means, standard deviations, and score ranges for the WJ-IV Broad Reading and Broad Written Language clusters, then conducted correlation analysis between writing and reading outcomes. To explore research question 2, which investigates variation in literacy scores across demographic groups, we calculated descriptive statistics for the WJ-IV Broad Reading and Broad Written Language clusters. For comparisons, we conducted independent samples *t*-tests and one-way ANOVAs, followed by post hoc comparisons where appropriate. We include a multiple regression analysis to determine how much of the variance in literacy outcomes are explained by demographic variables. For research question 3, examining how students’ literacy scores vary based on their language proficiency levels, we calculated descriptive statistics for WJ-IV clusters and conducted ANOVA with post hoc comparisons. Then, we conducted an analysis of covariance (ANCOVA) to determine whether differences in literacy outcomes across deaf family groups (caregivers, siblings, none) remain after adjusting for language proficiency. To address research question 4, which explores the extent to which language proficiency and phonological knowledge explain variance in literacy scores, we conducted a multiple linear regression. Predictor variables included language proficiency and PPVT GSV scores (serving as a proxy for phonological access, via sound or fingerspelling, to stored vocabulary). Models were estimated for WJ-IV Broad Reading and Broad Written Language outcomes using W scores.

## 3. Results

### 3.1. Research Question 1

What are the literacy scores of deaf elementary students in the US? What is the relationship between writing and reading outcomes?

Overall W scores and standard scores for the WJ-IV Broad Written Language and Broad Reading are reported in Table 3. Based on the written language scores for 3rd–6th grade students in this sample (n = 361), the data present as wide-ranging (40–119 standard score), with an overall mean of 68.32. According to the reported reading scores (n = 304), the data show similar patterns in range (40–122 standard score) and an overall mean of 65.11.

The WJ-IV Broad Written Language and Broad Reading have a normative mean of 100 and a standard deviation of 15. The number and percentage of standard scores above and below the normative mean can be viewed in Table 4. Approximately 23% of students scored within the average range or 1 SD above the normative mean in writing. Another 26% of students scored 1–2 SD below, and a little more than half the sample (51%) were 2 SD or more below the normative mean. In reading, slightly fewer students (21.1%) scored within the average range or 1 SD above the normative mean. Another 20.7% scored 1–2 SD below the mean, and 58.2% scored 2 SD or more below the normative mean.

To examine the relationship between students’ writing and reading outcomes, a Pearson correlation coefficient was calculated. The analysis revealed a significant positive correlation between the two variables, *r*(300) = 0.867, *p* < 0.001, indicating that students who performed well in writing also tended to perform well in reading. This result suggests a strong linear association between students’ literacy skills. A scatterplot was used to visualize the relationship between students’ reading and writing scores (Figure 1). Notably, the majority of data points fell above this line, suggesting that most students scored higher in writing than in reading.

### 3.2. Research Question 2

How do literacy scores vary across student demographic groups?

Literacy scores are presented by grade level in Table 5. The mean W scores between 3rd and 4th grade show increases in writing (from 448 to 461) and reading (from 427 to 442). Meanwhile, the mean standard scores for this time period remain in the low 70s, 1–2 SD below the mean. Between the 4th and 6th grades, the W scores largely remain unchanged, and the mean standard scores show a consistent drop in writing (from 73 to 64) and reading (from 70 to 58). Relative to the national norm (M = 100, SD = 15), the decline in standard scores by grade level suggests that the literacy performance of deaf students increasingly diverges from the norm as they progress through later elementary school. This widening gap indicates a growing disparity in literacy achievement compared to grade-level expectations.

In Table 6, Table 7 and Table 8, the writing and reading data are presented by gender, race, and hearing level. Hearing level accounts for the consistency of device use. For students who used their hearing technologies frequently to always (n = 209), amplified levels were used, and for those who did not use or have technologies, the unamplified hearing levels were used (n = 89). Minimal-to-slight variation was observed for the mean and range across categories, except in cases of insufficient numbers (e.g., two Native American students). Prior research has found gender, race, and hearing level to be predictive of literacy outcomes; so, we examined the relationship in this study. Two multiple regression analyses were conducted—one with writing outcomes and one with reading outcomes. The regression model for writing was significant, *F*(3, 352) = 2.94, *p* = 0.03. The model had an *R*^2^ value of 0.02, explaining approximately 2% of the variance in writing outcomes. Hearing level was a significant predictor (β = −0.12, *p* = 0.02), while gender (β = −0.03, *p* = 0.52) and race (β = −0.09, *p* = 0.08) were not. The regression model for reading was also significant, *F*(3, 295) = 2.72, *p* = 0.05, and explained approximately 3% of the variance in reading outcomes. Hearing level was a significant predictor (β = −0.13, *p* = 0.03), while gender (β = −0.05, *p* = 0.35) and race (β = −0.09, *p* = 0.11) were not. These results suggest that while hearing level is a significant predictor, it explains a tiny fraction of the variance in literacy outcomes.

A Bonferroni-adjusted alpha of 0.025 was applied to the next two analyses to control for family-wise errors. In Table 9, scores for the deaf elementary students are disaggregated by whether or not they have a disability. For 28% of students identified as having a disability in this sample, their literacy scores are consistently below students without an identified disability. Independent samples *t*-tests were conducted to compare the literacy scores between disabled and non-disabled students. After correcting for unequal variances, the results showed significant differences with moderate-to-large effects for writing, *t*(104.83) = 4.08, *p* < 0.001, *d* = 0.6, and reading, *t*(44.77) = 3.63, *p* < 0.001, *d* = 0.79.

The writing and reading scores are disaggregated by setting in Table 10. One-way ANOVAs were conducted to compare writing and reading outcomes across settings. The analyses revealed no significant differences between groups for writing, *F*(2, 358) = 1.59, *p* = 0.205, or reading, *F*(2, 301) = 0.405, *p* = 0.667. These findings, taken together with the high student mobility into schools for the deaf from other settings (as reported in the Method Section), suggest that setting must be studied under conditions of longevity during which the students have received consistent education in the settings.

In Table 11, literacy scores are disaggregated by whether the student has at least one deaf caregiver in the home, at least one deaf sibling, or no deaf family members in the home. A one-way ANOVA was conducted to examine the effect of a deaf caregiver or sibling on literacy performance. The results revealed a significant effect of small size on writing, *F*(2, 329) = 7.39, *p* < 0.001, η^2^ = 0.04, and of medium size on reading, *F*(2, 272) = 7.67, *p* < 0.001, η^2^ = 0.05. Post hoc comparisons using Tukey’s Honestly Significant Difference (HSD) indicated that students with a deaf caregiver in the home scored significantly higher in writing than those with a deaf sibling, *p* = 0.007, and those without a deaf person in the home, *p* = 0.001. Similarly, those with a deaf caregiver scored significantly higher in reading than those with a deaf sibling, *p* = 0.037, and those without a deaf person in the home, *p* < 0.001. There were not significant differences between students with deaf siblings and those without.

### 3.3. Research Question 3

How do students’ literacy scores vary based on language proficiency levels?

The following tables (Table 12, Table 13 and Table 14) present the literacy data disaggregated by language proficiency categories. Table 12 presents writing and reading scores for those with an ASL proficiency rating. Table 13 includes students using spoken English. Bilingual, bimodal students are represented in both of these tables, using their distinct proficiency ratings for ASL or spoken English. Lastly, Table 14 displays the literacy outcomes by language proficiency, which is a variable representing students’ top proficiency level in either ASL or spoken English. Across all tables and regardless of which language, there is a clear upward trend in writing and reading means (both W and standard scores) as the students’ language proficiency level increases. This trend is additionally illustrated in violin plots by language proficiency rating in Figure 2. Furthermore, Figure 2 demonstrates that at every language proficiency level, writing scores outpace reading scores.

ANOVA was conducted to compare the effect of language proficiency on literacy outcomes. Significant differences were found for writing, *F*(4, 347) = 28.97, *p* < 0.001, η^2^ = 0.25, with eta-squared indicating a large effect size. Post hoc comparisons using Tukey’s HSD revealed significant differences between every language rating. Those rated as expressing most anything in at least one language scored significantly higher on writing than those who expressed many things, *p* < 0.001; those having difficulty expressing many things, *p* < 0.001; and those having difficulty expressing most things, *p* < 0.001. Those rated as expressing many things in at least one language scored significantly higher than those having difficulty expressing many things, *p* = 0.019, and those having difficulty expressing most things, *p* < 0.001. Those rated as having difficulty expressing many things in a language scored significantly higher on writing than those having difficulty expressing most things, *p* < 0.001.

Similar ANOVA outcomes were found in comparing the effect of language proficiency on reading, *F*(4, 290) = 26.53, *p* < 0.001, η^2^ = 0.27. Again, post hoc comparisons using Tukey’s HSD revealed significant differences between every language rating. Those rated as expressing most anything in at least one language scored significantly higher on reading than those who expressed many things, *p* < 0.001; those having difficulty expressing many things, *p* < 0.001; and those having difficulty expressing most things, *p* < 0.001. Those rated as expressing many things in at least one language scored significantly higher than those having difficulty expressing many things, *p* = 0.018, and those having difficulty expressing most things, *p* < 0.001. Those rated as having difficulty expressing many things in a language scored significantly higher on writing than those having difficulty expressing most things, *p* = 0.026.

The standard scores for students with the highest language proficiency rating range from 40 to 120, with means of 77 (writing) and 75 (reading). Among those who have the most difficulty expressing themselves in a language, the range is more restricted, from 40 to 82, with means of 49 (writing) and 46 (reading), more than 3 SD below the normative mean. Upon a reviewer’s request, we disaggregated data by students who received the highest proficiency rating for both ASL and spoken English (n = 22). There were no noticeable differences in the means and ranges for this subgroup as compared to the 136 students reported in Table 14 with the highest proficiency rating in at least one language.

Next, an analysis of covariance (ANCOVA) was conducted to determine whether differences in literacy outcomes across deaf family groups (caregivers, siblings, none) remained after adjusting for language proficiency. The overall model for writing was significant, with language proficiency emerging as a significant covariate, *F*(1, 328) = 59.97, *p* < 0.001, η^2^ = 0.16. After controlling for language proficiency, the effect of deaf family groups on literacy outcomes was no longer significant, *F*(2, 328) = 2.21, *p* = 0.11. Similarly, the overall model for reading was significant, with language proficiency as a significant covariate, *F*(1, 271) = 79.51, *p* < 0.001, η^2^ = 0.23. However, after controlling for language proficiency, the effect of deaf family groups on literacy outcomes was no longer significant, *F*(2, 271) = 0.74, *p* = 0.48. These findings indicate that language proficiency plays a critical role in shaping literacy outcomes for deaf students, rendering the influence of family group differences insignificant once controlled for.

### 3.4. Research Question 4

To what extent do language proficiency and phonological knowledge (spoken or fingerspelled) explain variance in students’ literacy scores?

Our analysis sought to examine the relationship between deaf students’ language proficiency and phonological knowledge and their literacy outcomes. Two multiple regression analyses were conducted to examine whether language proficiency and phonological knowledge predicted writing and reading outcomes. The regression model for writing was significant, *F*(2, 294) = 179.3, *p* < 0.001. The model had an *R*^2^ value of 0.55, explaining approximately 55% of the variance in writing outcomes. Both language proficiency (*B* = 9.43, SE = 2.38, β = 0.17, *p* < 0.001) and phonological knowledge (*B* = 0.48, SE = 0.03, β = 0.65, *p* < 0.001) were significant predictors. The regression model for reading was also significant, *F*(2, 240) = 200.59, *p* < 0.001, and explained approximately 63% of the variance in reading outcomes. Both language proficiency (*B* = 16.72, SE = 3.25, β = 0.23, *p* < 0.001) and phonological knowledge (*B* = 0.65, SE = 0.04, β = 0.66, *p* < 0.001) were significant predictors. In both models, variance inflation factor (VIF) values were 1.2, indicating no multicollinearity issues among the two predictors. These results suggest that higher levels of both language proficiency (whether ASL and/or spoken English) and phonological knowledge (fingerspelled and/or spoken) are predictive of better literacy outcomes. Additionally, the standardized coefficients associated with both models indicate a greater effect on literacy outcomes due to phonological knowledge than language proficiency.

## 4. Discussion

Previous studies have identified variables as predictors of literacy development, including type of hearing loss (e.g., unilateral versus bilateral loss; [46]), aided hearing as a factor of speech intelligibility ([64]), race ([50]), and gender ([49]). The current study provides the largest and most recent sample of students receiving deaf education services in later elementary grades. It is inclusive of students with mild-to-profound hearing levels and spoken and/or sign language modalities. Race and gender were not significant predictors in this study. Hearing level was significant but accounted for only 2–3% of the variation in the regression models. Rather, language proficiency, regardless of modality, emerged as a strong predictor of reading and writing performance. Students who demonstrated higher proficiency in spoken English or ASL had significantly higher literacy scores on the WJ-IV Broad Written Language and Broad Reading measures, reiterating the power and role of an accessible, strong language foundation. Prior research has shown that strong language proficiency, whether in spoken or signed language, is necessary for complex literacy development among deaf students ([40]). Findings from this study align with and reinforce this relationship. Students with stronger expressive language skills had consistently higher scores.

Another well-documented predictor of literacy performance is the presence of a deaf caregiver in the home ([1]; [22]). It makes sense that having a signing deaf caregiver and early exposure to ASL are correlated and that this interaction variable has proven to be a significant predictor of reading comprehension ([59]). Our results echoed these findings, showing that students with a deaf caregiver scored significantly higher in both reading and writing than those with a deaf sibling or no deaf person in the home. However, when we controlled for language proficiency, these group differences were no longer statistically significant. This suggests that it is not only the caregiver’s hearing status that influences literacy outcomes; early and consistent exposure to accessible language is key. This finding aligns with recent research showing that the language and literacy gains typically associated with deaf signing families can extend to hearing families who adopt accessible, visual language practices ([2]; [10]; [21]).

The strength of language proficiency in explaining literacy performance is encouraging; however, persistently low mean scores, even after disaggregating for disability, point to a broader concern. The mean literacy scores in this sample were below average, with most students scoring 1–3 standard deviations below the mean and approximately a third of the sample scoring more than 3 standard deviations below the mean (between 40 and 55 standard score). As a whole, these data present more than individual variation. They reflect the enduring impacts of environments in which many deaf children are raised and educated. Language deprivation, which is the lack of early and consistent exposure to a fully accessible language, has lasting impacts on academic trajectories long after the beginning years of life. Without access to quality language during this time, children are at risk for delays in their development that have consequences for syntactically complex language acquisition ([13]), literacy ([36]), and cognition ([32]). These findings stress the importance of supporting families from the earliest opportunity to prioritize environments with rich and accessible language.

Careful analysis of our data additionally suggests that while accessible language and subsequent proficiency lay the groundwork, it is through intentional, responsive instruction that students build strong literacy skills from that foundation. Although language proficiency accounted for substantial variance in literacy scores, phonological representations of vocabulary (spoken or fingerspelled) were the strongest predictor. These results highlight that language proficiency is a critical factor in explaining literacy outcomes; but it cannot tell the whole story. A positive trajectory depends on instructional approaches that align with students’ language strengths and needs. In other words, instruction should be accessible ([31]) and also work to expand students’ communicative resources through translanguaging strategies and linguistically responsive instruction ([18]; [71]). There are evidence-based practices that align with and support the literacy development of deaf students ([73]), whether their cognitive linguistic systems are based in visual and/or spoken language ([4]). A strong match, characterized by alignment between a student’s language use and their teachers’ language and literacy instruction, provides optimal conditions for meaningful and consistent growth. These findings highlight the need for future research to investigate the inner workings of instructional models aimed at widening students’ linguistic abilities to perceive and express language effectively, particularly those implemented in environments supporting a diverse range of languages and modalities.

### 4.1. Higher Writing Scores than Reading

Overall, writing scores in our sample were consistently higher than reading scores. [64] ([64]) reported a similar finding based on a sample of monolingual, hard-of-hearing elementary students. This pattern suggests the potential importance of leveraging writing as a tool to spur reading development ([17]; [61]). Writing instruction that involves active, purposeful communication directed toward specific audiences and for multiple purposes gives students agency in their compositions ([39]) and boosts the motivation to engage in challenging learning ([43]). Students typically draw on their known linguistic resources to generate authentic messages, fostering meaningful engagement ([16]). This productive, context-rich use of language contrasts with reading, which is receptive and more decontextualized from personal experience. Reading requires the individual to decode and comprehend language that is not self-generated but presented by the author, thus affording less personal control and contextual grounding. Supporting writing as a bridge to reading may therefore provide a more accessible and motivating pathway for deaf learners to develop literacy skills ([55]).

Another consideration in interpreting the differences in writing and reading outcomes is the possibility that assessment measures may differently impact signing deaf students. The letter–word identification subtest of the WJ-IV Broad Reading cluster, for example, assesses not only the ability to identify letters and words but also implicitly taps vocabulary knowledge when students decode and then express the words through sign. A signed response from students indicates they have successfully decoded the word and retrieved it from their cognitive bank. For students responding using English, their responses are spoken aloud and do not necessarily indicate that they know the word’s meaning. In this study, administration instructions were adapted to allow signing students to fluently fingerspell words; however, habits to respond through sign language may have prevailed. Responding with the signed vocabulary words arguably would make this subtest more challenging than the experience of hearing students (upon which it was normed), and it could potentially explain the lower scores in reading compared to writing. Given the diverse linguistic backgrounds of deaf students, including various combinations of modality and language, assessments normed on hearing monolingual populations often fail to capture their full range of knowledge and skills. There is a need for assessments specifically designed for and normed on deaf students to ensure valid and equitable measurement of literacy ([37]; [62]).

### 4.2. Progression Through the Elementary Grades

Data from this study indicate that by third grade, students’ average reading and writing scores are one standard deviation below the normative mean. Across subsequent grade levels, students are not progressing quickly enough to maintain or improve their standing. By sixth grade, the gap widens to 2–3 standard deviations below the normative mean. This pattern builds from the findings of [3] ([3]), who investigated the reading skills of deaf students in kindergarten through second grade. They found that the students made expected gains in language structures and early vocabulary, but reading skills, particularly comprehension, declined by grade level. Together, these results from national datasets of kindergarten through sixth-grade performance suggest that early gains in language and literacy do not automatically translate into sustained growth without effective, targeted support. In contrast, studies with even younger children capture a more hopeful starting line. [2] ([2]) investigated early reading trajectories in a large sample of preschool deaf children and found that, regardless of home language background, many demonstrated consistent growth in word identification and early alphabetic skills. The most prominent variable explaining these gains was early, consistent exposure to accessible visual language. A prior study also found receptive ASL skills and fingerspelling proficiency to strongly predict emergent alphabetic knowledge in preschoolers ([1]). These findings collectively point to the importance of differentiation and instructional alignment as students move into later grades, where literacy demands change. Without targeted and sustained support, early gains may be difficult to maintain, particularly as reading and writing tasks become more complex and require higher-level comprehension and language skills.

### 4.3. Educational Setting

General claims about the language and literacy development of deaf students, based on educational settings or program philosophies, must be made with extreme caution, as placement context is inextricably intertwined with other variables, such as age of entry, consistency of placement, and instructional variability. Even when academic outcomes appear to differ across settings, these differences are often shaped by antecedents or broader contextual factors. For example, [67] ([67]) found that deaf students with cochlear implants in the Netherlands who began in special schools and later transitioned to mainstream settings outperformed those who had been consistently educated in mainstream classes on government required tests. While this may suggest early specialized instruction from special schools made the difference, socioeconomic status was a potentially confounding variable. Other studies emphasize the importance of early, high-quality, and sustained access to language for later cognitive and academic outcomes. [34] ([34]) found that deaf students who entered a bilingual school for the deaf earlier demonstrated stronger proficiency in ASL syntax and analogical reasoning. Similarly, [21] ([21]) found that deaf students with hearing caregivers who had earlier enrollment in bilingual programs experienced academic outcomes comparable to those of deaf students with deaf caregivers, emphasizing the importance of early, accessible instruction that matches students’ linguistic strengths and needs.

### 4.4. Limitations and Future Research

The sample in this study included deaf elementary students who received language and literacy instruction from a teacher of deaf students (TOD) for a minimum of 2 h a week in either self-contained or itinerant settings. Deaf students who received less than 2 h a week or no support from a TOD were not included in this study, and therefore, conclusions may not be drawn about this group.

Second, data were collected on whether there was a deaf caregiver or sibling in the home. While students with a deaf caregiver in the home show significantly higher literacy outcomes to students without a deaf caregiver, these effects were not maintained when language proficiency was factored into the model. Recent studies demonstrate that factors such as hearing parents’ learning of sign language upon having a deaf child ([10]) and/or early enrollment of deaf children into bilingual education programs ([21]) can have an equally positive impact on language and literacy outcomes as having deaf parents. Additionally, since having a deaf caregiver in the home does not automatically guarantee quality language access, future studies should include information on parents’ sign language proficiency and students’ early interventions and schooling experiences.

Third, one reviewer raised concerns regarding how hearing levels were measured. Hearing thresholds (both unaided and aided) were obtained from students’ audiograms provided by school programs. The reviewer noted that aided soundfield audiograms can be limited due to hearing aid processing algorithms—such as nonlinear gain that amplifies soft and loud sounds differently—making them an imperfect representation of functional hearing. While this study accounts for the consistency of device use, we did not have access to data on the duration of device use or the quality of device fitting, both of which may influence outcomes. Therefore, hearing levels in this study should be viewed as a general estimate of auditory access.

The high degree of transience among deaf students in this study highlights the difficulty of drawing concrete conclusions about the influence of educational setting or program communication philosophy on performance. Of the students for whom placement data were available (n = 286), nearly half (49%) had transferred between schools or programs, most commonly moving from public school settings using total communication, listening/spoken language, or sign-supported speech to schools for the deaf with bilingual philosophies (71%). These transitions often occur after it becomes clear that language and literacy growth are not progressing as expected, which complicates analyses that attempt to link performance to current placement or philosophy. Students who transfer in later grades may still be experiencing the effects of earlier language deprivation ([34]) or ineffective instruction ([11]). Even students who remain in one setting may experience inconsistency in the instructional approach and specialized support, especially those with additional disabilities, as shifts in educational classification can impact service delivery ([5]). As such, conclusions about educational setting or communication approach must be made cautiously, with attention to students’ full educational trajectories. Analyses that fail to account for the timing and nature of educational transitions risk inaccurately attributing outcomes to the current setting and may disproportionately and unfairly reflect on schools for the deaf or bilingual education models. Future research should prioritize longitudinal designs and focus on students with stable placement histories to more accurately assess how specific environments support literacy development over time.

Finally, while this study reinforces that language proficiency is a predictor of literacy outcomes, it does not fully explain the variability observed across the sample. When language proficiency is observed alongside phonological representations of vocabulary, particularly when aligned with students’ expressive language (signed or spoken), our model accounts for 55–63% of the variance in literacy performance. These findings underscore the importance of both accessible language and instructional alignment. Future research should examine the intersection of language and literacy instruction, particularly among students who perform within or above the expected range. These studies could explore the ways in which instructional approaches are aligned (or misaligned) with the language used by students and the extent to which that alignment supports literacy development. This line of inquiry is important for advancing evidence-based practices that equitably serve the diverse strengths and needs of deaf learners.

## 5. Conclusions

Utilizing large-scale datasets helps in identifying patterns and interactions between various learner variables and standardized measures. This study adds to the growing body of research on literacy development among deaf students by demonstrating that language proficiency, regardless of modality, is more predictive of reading and writing outcomes than demographic variables. The observed pattern of higher writing scores relative to reading suggests an underexplored instructional opportunity to use writing as a pathway to strengthen reading development. The documented widening gap in literacy performance across grades underscores the need for sustained, targeted instructional support aligned with students’ language strengths. Together, these findings deepen our understanding of the trajectories and mechanisms of literacy among deaf students and highlight the importance of linguistically responsive instruction and equitable assessment practices.

## Figures and Tables

**Figure 1 behavsci-15-01100-f001:**
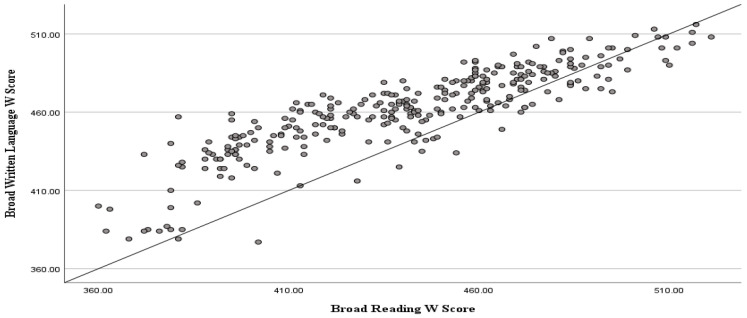
Scatterplot of the relationship between reading and writing scores. Note: Reading scores are displayed on the *x*-axis and writing scores on the *y*-axis. A line of equality (y = x) is included to indicate where scores in both areas would be equal.

**Figure 2 behavsci-15-01100-f002:**
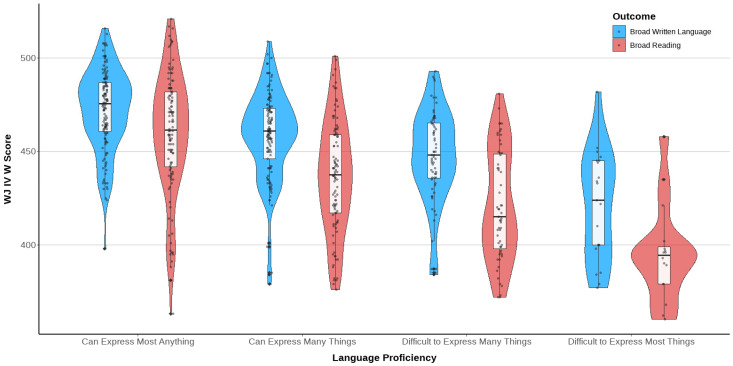
Violin plots of W scores by language proficiency levels.

**Table 1 behavsci-15-01100-t001:** Participant demographics (n = 368).

Demographic	Category	n	%
Grade Level	Grade 3	69	18.8
	Grade 4	96	26.1
	Grade 5	104	28.3
	Grade 6	97	26.4
	Not reported	2	0.5
Gender	Female	158	42.9
	Male	206	56.8
	Not reported	1	0.3
Race/Ethnicity	African American	88	23
	Asian or Pacific Islander	21	5.7
	Latinx	82	22.3
	Multiracial	19	5.2
	Native American	2	0.5
	White	140	38
	Not Reported	3	0.8
	Other	12	3.3

Note: Percentages are based on the total sample (N = 368), though not all participants responded to every item.

**Table 2 behavsci-15-01100-t002:** Teacher ratings of students’ proficiency in ASL and spoken English on a scale of 1–5 (n = 368).

Proficiency Rating	ASL	Spoken English
Can express most anything (1)	100	64
Can express many things (2)	94	95
Difficulty expressing many things (3)	70	63
Difficulty expressing most things (4)	32	30
Does not use the language (5)	63	107
Not reported	9	9

**Table 3 behavsci-15-01100-t003:** Overall literacy scores.

WJ-IV Broad Written Language					WJ-IV Broad Reading				
	W Score		Standard Score			W Score		Standard Score	
*n*	*M (SD)*	Min–Max	*M (SD)*	Min–Max	*n*	*M (SD)*	Min–Max	*M (SD)*	Min–Max
361	459.92 (26.95)	377–516	68.32 (19.34)	40–119	304	440.88 (35.66)	360–521	65.11 (20.6)	40–122

**Table 4 behavsci-15-01100-t004:** Number and percentage of literacy scores above or below the normative mean.

		WJ-IV Broad Written Language		WJ-IV Broad Reading	
Category	Range	*n*	*%*	*n*	*%*
1–2 SD Above Mean	115–130	1	0.30%	5	1.70%
Average Range	85–115	82	22.70%	59	19.40%
1–2 SD Below Mean	70–85	94	26%	63	20.70%
2–3 SD Below Mean	55–70	80	22.20%	70	23%
3+ SD Below Mean	40–55	104	28.80%	107	35.20%

**Table 5 behavsci-15-01100-t005:** Literacy scores by grade level.

	WJ-IV Broad Written Language					WJ-IV Broad Reading				
		W Score		Standard Score			W Score		Standard Score	
Grade	*n*	*M (SD)*	Min–Max	*M (SD)*	Min–Max	*n*	*M (SD)*	Min–Max	*M (SD)*	Min–Max
3rd	66	447.52 (25.97)	379–493	72.18 (19.28)	40–112	52	427.46 (29.51)	372–481	72.37 (17.59)	40–122
4th	94	460.78 (25.93)	377–507	72.87 (18.33)	40–112	84	442.35 (36.71)	368–516	70.33 (21.31)	40–120
5th	103	462.25 (26.04)	387–509	65.77 (18.81)	40–110	77	443.4 (36.06)	360–521	62.48 (19.71)	40–117
6th	96	464.58 (27.46)	384–516	63.79 (19.84)	40–119	89	444.55 (36.53)	362–517	58.2 (20.14)	40–120

Note: Data were collected at the start of the school year.

**Table 6 behavsci-15-01100-t006:** Literacy scores by gender.

	WJ-IV Broad Written Language					WJ-IV Broad Reading				
		W Score		Standard Score			W Score		Standard Score	
Gender	*n*	*M (SD)*	Min–Max	*M (SD)*	Min–Max	*n*	*M (SD)*	Min–Max	*M (SD)*	Min–Max
Female	154	461.16 (26.44)	377–511	68.74 (19.98)	40–119	122	443.52 (35.19)	360–516	67.04 (20.43)	40–120
Male	206	458.91 (27.38)	379–516	68.01 (18.95)	40–112	181	438.98 (36.02)	362–521	63.85 (20.72)	40–122

**Table 7 behavsci-15-01100-t007:** Literacy scores by race.

	WJ-IV Broad Written Language					WJ-IV Broad Reading				
		W Score		Standard Score			W Score		Standard Score	
Race	*n*	*M (SD)*	Min–Max	*M (SD)*	Min–Max	*n*	*M (SD)*	Min–Max	*M (SD)*	Min–Max
White	137	463.12 (26.11)	384–516	72.28 (20.34)	40–119	126	445.42 (35.31)	362–517	69.06 (21.91)	40–122
Black/African American	87	462.76 (26.42)	377–508	69.66 (17.26)	40–108	75	441.53 (36.33)	372–521	64.77 (19.91)	40–117
Latine	80	452.56 (26.25)	379–513	62.55 (18.64)	40–107	57	433.81 (31.71)	363–506	61.33 (18.23)	40–102
Asian/Pacific Islander	21	455.19 (29.63)	384–507	63 (19.48)	40–100	15	428 (39.14)	376–494	56.13 (17.74)	40–92
Multiracial	18	468.67 (27.66)	416–509	70.5 (21.12)	40–104	17	451.18 (38.19)	396–509	64.71 (22.91)	40–102
Other	13	450.85 (27.08)	379–488	60.69 (14.48)	40–85	9	429.33 (32.9)	368–485	57.78 (15.83)	40–79
Native American	2	418 (25.46)	400–436	46.5 (9.19)	40–53	2	374 (19.8)	369–388	40 (0)	40–40

**Table 8 behavsci-15-01100-t008:** Literacy scores by hearing level.

	WJ-IV Broad Written Language					WJ-IV Broad Reading				
		W Score		Standard Score			W Score		Standard Score	
Hearing Level	*n*	*M (SD)*	Min–Max	*M (SD)*	Min–Max	*n*	*M (SD)*	Min–Max	*M (SD)*	Min–Max
0–25 dB	52	462.92 (25.14)	379–508	70.02 (19.46)	40–112	42	443.9 (33.96)	381–516	64.86 (20.47)	40–120
26–40 dB	56	463.09 (25.31)	384–511	71.43 (20.14)	40–119	47	451 (33.32)	362–516	72.83 (20.84)	40–122
41–55 dB	32	469.44 (29.18)	400–516	75.69 (18.85)	40–109	27	450.52 (43.37)	360–517	71.48 (21.16)	40–107
56–70 dB	27	461.74 (27.2)	384–501	70.04 (17.81)	40–97	20	436.45 (41.86)	372–512	64 (22.41)	40–103
71–90 dB	30	461.33 (23.51)	387–501	69.43 (17.44)	40–110	24	439.88 (31.22)	378–508	64.88 (17.81)	40–115
91+ dB	95	450.31 (28.95)	377–508	60.87 (18.1)	40–111	82	427.02 (33.31)	363–509	56.95 (17.41)	40–105
Not Available	64	461.41 (24.62)	400–508	69.52 (20.01)	40–108	57	446.47 (33.78)	372–521	67.88 (22.23)	40–117

Note: Amplified levels were used for students who wear their hearing technologies frequently to always.

**Table 9 behavsci-15-01100-t009:** Literacy scores by disability.

	WJ-IV Broad Written Language					WJ-IV Broad Reading				
		W Score		Standard Score			W Score		Standard Score	
	*n*	*M (SD)*	Min–Max	*M (SD)*	Min–Max	*n*	*M (SD)*	Min–Max	*M (SD)*	Min–Max
No Disability	278	463.13 (24.52)	379–516	70.16 (18.46)	40–112	236	444.65 (33.65)	362–521	66.64 (19.9)	40–120
Disability	78	447.45 (31.36)	377–513	60.71 (20.09)	40–112	63	425.43 (38.69)	360–509	58.32 (21.32)	40–122

**Table 10 behavsci-15-01100-t010:** Literacy scores by setting.

	WJ-IV Broad Written Language					WJ-IV Broad Reading				
		W Score		Standard Score			W Score		Standard Score	
Setting	*n*	*M (SD)*	Min–Max	*M (SD)*	Min–Max	*n*	*M (SD)*	Min–Max	*M (SD)*	Min–Max
Deaf School	215	461.03 (26.93)	377–516	67.42 (19.68)	40–119	188	441.99 (35.12)	362–517	63.99 (20.32)	40–120
Public School	126	459.63 (26.43)	387–509	69.96 (18.23)	40–111	96	439.94 (35.75)	360–521	66.46 (20.43)	40–117
Itinerant	20	449.85 (29.53)	379–508	67.75 (22.7)	40–112	20	434.9 (41.2)	372–516	69.15 (24.06)	40–122

Note: Public school represents students receiving literacy instruction from a deaf education teacher in a self-contained classroom.

**Table 11 behavsci-15-01100-t011:** Literacy scores by deaf family members.

	WJ-IV Broad Written Language					WJ-IV Broad Reading				
		W Score		Standard Score			W Score		Standard Score	
Deaf Member	*n*	*M (SD)*	Min–Max	*M (SD)*	Min–Max	*n*	*M (SD)*	Min–Max	*M (SD)*	Min–Max
Deaf caregiver	60	471.72 (25.57)	385–516	79.87 (17.56)	40–112	58	456.86 (34.75)	379–517	76.17 (19.94)	40–122
Deaf sibling	29	453.83 (30.07)	384–500	66.66 (19.42)	40–109	23	435.91 (33.4)	372–499	64.43 (20.7)	40–107
None	243	458.22 (25.77)	377–509	66.19 (18.6)	40–111	194	437.16 (34.26)	362–521	62.69 (19.61)	40–117

Note: Some homes with deaf caregivers also had deaf siblings. Those marked as having deaf siblings had no deaf caregivers in the home.

**Table 12 behavsci-15-01100-t012:** Literacy scores by ASL proficiency.

	WJ-IV Broad Written Language					WJ-IV Broad Reading				
		W Score		Standard Score			W Score		Standard Score	
Proficiency	*n*	*M (SD)*	Min–Max	*M (SD)*	Min–Max	*n*	*M (SD)*	Min–Max	*M (SD)*	Min–Max
Can express most anything	97	470.43 (21.57)	398–516	78.86 (18.05)	40–111	82	454.5 (32.19)	363–517	71.7 (19.33)	40–115
Can express many things	93	457.53 (23.19)	384–504	65.63 (18.85)	40–112	77	436.47 (31.08)	376–516	61.08 (21.07)	40–122
Difficult to express many things	69	452.43 (28.22)	384–508	61.28 (18.43)	40–108	58	427.26 (35.81)	372–521	56.09 (19.01)	40–117
Difficult to express most things	30	433.73 (35.23)	377–501	53.03 (17.72)	40–101	22	417.59 (43.75)	360–494	55.91 (19.42)	40–96

**Table 13 behavsci-15-01100-t013:** Literacy scores by spoken English proficiency.

	WJ-IV Broad Written Language					WJ-IV Broad Reading				
		W Score		Standard Score			W Score		Standard Score	
Proficiency	*n*	*M (SD)*	Min–Max	*M (SD)*	Min–Max	*n*	*M (SD)*	Min–Max	*M (SD)*	Min–Max
Can express most anything	64	477.17 (21.88)	430–516	81.64 (18.16)	40–112	56	470.82 (29.6)	391–521	80.77 (19.86)	40–120
Can express many things	93	465.24 (21.9)	379–509	72.78 (17.23)	40–112	73	447.05 (29.48)	381–501	69.29 (19.62)	40–122
Difficult to express many things	61	454.1 (23.97)	384–493	62.93 (15.71)	40–89	53	431.26 (31.55)	363–510	59.85 (15.79	40–101
Difficult to express most things	29	452.24 (20.02)	401–482	59.59 (14.24)	40–82	23	427.3 (29.68)	379–486	55.52 (15.29)	40–89

**Table 14 behavsci-15-01100-t014:** Literacy scores by language proficiency.

	WJ-IV Broad Written Language					WJ-IV Broad Reading				
		W Score		Standard Score			W Score		Standard Score	
Proficiency	*n*	*M (SD)*	Min–Max	*M (SD)*	Min–Max	*n*	*M (SD)*	Min–Max	*M (SD)*	Min–Max
Can express most anything	136	472.45 (21.6)	398–516	77.32 (18.22)	40–112	116	459.66 (32.58)	363–521	75.39 (20.14)	40–120
Can express many things	125	458.22 (23.58)	379–509	66.37 (17.56)	40–112	102	436.51 (30.01)	376–501	61.69 (19.06)	40–122
Difficult to express many things	68	447.35 (25.08)	384–493	57.81 (15.61)	40–89	59	420.93 (29.44)	372–481	54.25 (14.48)	40–87
Difficult to express most things	21	421.29 (28.95)	377–482	48.81 (12.8)	40–82	16	395.19 (25.73)	360–458	45.94 (10.32)	40–71

Note: Based on students’ top proficiency rating in either language.

## Data Availability

The data will be publicly available at this DOI (https://doi.org/10.5281/zenodo.16420315 accessed on 1 January 2027) following completion of the project’s final analyses.

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
