# Peer review of "Variability in Language and Literacy Outcomes Among Deaf Elementary Students in a National Sample"

_behavsci, 2025, doi:10.3390/bs15081100_

Round 1
Reviewer 1 Report
Comments and Suggestions for Authors
Dear authors,
Thank you so much for this important and interesting study on the variability in language and literacy outcomes among deaf students. Studies with larger samples like these are a goldmine and an important contribution to the field. I would like to give some suggestions to further strengthen your manuscript and maybe even get more from your data than you did now.
Literature review:
In the introduction of your manuscript you give a general overview of the heterogeneity in the sample of deaf and hard of hearing (DHH) children and the impact this may have on their reading and writing development. You discuss topics such as early exposure to language, full access to language and the role of sign language (interfering or not). However, your literature review does not focus on what we know from previous research about the impact of the factors in the reading and writing development of DHH children, the topic of your study. Without a more thorough discussion of these factors, it is not clear why you decide to look at the impact of hearing thresholds, gender, race as factors in the variability of literacy scores and the impact of language proficiency and phonological knowledge in research questions 3 and 4. On page 12, line 455, you mention that prior research has found gender, race, and hearing level to be predictive of literacy outcomes. I would like to read about that literature in your introduction. What factors does the literature point to?
Research questions: I would suggest to move the research questions to the end of the introduction section instead of the beginning of the methods section.
Methods section
Participants
Your methods section provides a lot of demographic information about the participants in the text and in tables. That seems somewhat redundant. I would suggest to either delete Tables 1 and 2 or choose to delete the information from the text and include one large table with all the demographic information.
I was wondering if you could mention something about the representativeness of your sample as they are not evenly distributed over the educational settings. Is this representative of the DHH population in the US? Please add something about this with your report of educational settings.
In discussing the educational settings, you mention the high degree of student mobility in this respect. I would like to point out that Van der Straaten et al. (2021) found that the group of students with CI who started in deaf education and then transferred to mainstream education outperformed DHH students in mainstream education who did not transfer between schools. This might be interesting to include in your discussion.
In reporting the hearing levels of children, you mention that 8 students had no hearing loss (0-25 dB). Why did you not exclude these children from your data set? One of the strong characteristics of your study is the comparison within the DHH group and I would advise to avoid all noise here by deleting the students within the 0-25 dB range.
It is not entirely clear why you report both the hearing levels without and with amplification. Could you please clarify, also which measure you eventually use in your analyses?
Language proficiency: this is measured through teacher report. Could you please add whether these teachers were fluent in the languages themselves. Were they able to assess both languages in the students? Were they fluent in both languages? And if not, how did they assess student proficiency?
As for the language proficiency, I think it would be very interesting to also provide the number of students who were proficient in both languages and also use this group in the analyses (I will get back to that when discussing the results).
Measures
PPVT: it is not entirely clear how the PPVT was administered. At first you mention that no additional signs were provided, but then (page 9, lines 368-369) you mention that for students who used both ASL and English, both modalities were provided. Does this mean that both the spoken word and the sign were provided during administration? The PPVT is a measure for spoken vocabulary. When you add ASL, you are measuring ASL vocabulary. Can you please clarify and also mention whether you were able to distinguish the scores within and between children for ASL and English vocabulary?
Then you explain that the administration approach made it possible to use this measure as a proxy for phonological knowledge. This needs further explanation for it is now unclear how this exactly worked and why you did not include a measure of phonological knowledge and fingerspelling knowledge. It is now unclear how you came to scores for these variables and how you distinguished between the two.
Analyses: in describing the analyses for research question 3 suddenly the role of deaf family members comes up. This was not a part of the research question and if you would like to study the difference between children with and without deaf family members it would better fit research question 2 on demographics than question 3. Also, having a deaf caregiver is not the same as growing up with sign language (which is also important to point to in your discussion section). It is unclear why you include this here which also comes back to discussing the literature on factors in DHH children's literacy development in the introduction.
Results section
This section is very accessible. I have some suggestions to further improve it.
In answering research question 1, you describe the group mean. This is the mean of standard scores. From there you continue to discuss how many students score above the mean, average, or below the mean. The terminology is not entirely clear here, because 'mean' means something different here than the group mean. In Table 1 it becomes clear that here you compare children to the standard score, the norm, but this is not clear from the text. Please clarify this for the reader. In discussing the results of research question 2, you use the term 'national norm', that might be a solution here too.
The results of research question 2 also discuss the impact of hearing level. However, it is unclear whether you use the thresholds with or without amplification here (in table 8 there are suddenly 52 children with thresholds below 25 dB). Please clarify, also why you use one or the other. Also, I would again suggest to delete the children with thresholds below 25 dB as they pollute your sample. I would also like to suggest to report post hoc analyses to see which levels differ from each other: does it still matter at the higher thresholds?
For research question 3 you only report descriptive results. Please also provide statistical analyses on the differences between the groups. Are the differences between the groups significant and if so, between which groups? And are there differences between children who are proficient in ASL and those who are proficient in English? Also, I really think it would be interesting to add the comparison between children who are proficient in both languages versus those who are proficient in only one language. This would add to the discussion of bilingual education.
As for research question 4, I was wondering whether you could disentangle the role of fingerspelling and spoken phonological knowledge? As I mentioned before it is not clear how this is measured exactly which makes it difficult to interpret the findings discussed here.
Discussion
The discussion gives a nice overview of the findings and reflection on these findings. However, I miss a conclusion on what your study adds to the current state of knowledge. Have we now learned more about the factors in literacy development or have we learned more about the reading and writing trajectories of DHH children? In your discussion I would focus on what your study contributes to scientific knowledge.
Author Response
Dear Reviewers,
Thank you for your quality contributions and the opportunity to improve our manuscript. We have read your comments carefully and responded to every point of feedback (in table form, separated by reviewer number).
With kind regards,
Authors

Reviewer 2 Report
Comments and Suggestions for Authors
This is a well written article on an important topic. My primary concern is that the authors appear to be overgeneralizing the research done on children who are deaf to children who are hard of hearing. They did not take into account recent work that examines both of these groups, particularly papers by Werfel and Lund's research team on the ELLA study (which has published a number of papers on literacy) and Bruce Tomblin's papers on literacy from the Outcomes of Children with Hearing Loss study. These are two different longitudinal studies on a cohort of children who have had access to early identification and early intervention and it would be worth taking a look at those findings and placing the current results within that context.
My major concern with this paper is the authors' conclusion that hearing levels only contributed a small amount of variance to literacy. I don't disagree that language proficiency is an important predictor - my concern is how the authors went about measuring levels. It appears that they used aided soundfield audiograms to calculate hearing levels, although this isn't very clearly specified in the methods. This approach is flawed due to the processing algorithms of hearing aids (briefly - measuring aided thresholds isn't a valid approach to capture hearing levels with a hearing aid because of the nonlinear gain in the devices means that soft levels of sound are going to amplified at a different level than conversational or loud speech). The authors' approach also doesn't capture the amount of time children have been wearing their hearing aids. There isn't a simple one-to-one correspondence with aided hearing (or unaided hearing, for that matter) and language and literacy outcomes - it's moderated by factors related to the quality of the device fitting and the consistent of device use - and the authors appear to be oversimplifying this relationship, and therefore dismissing the role of aided hearing (again, I refer the authors' to papers by Bruce Tomblin and Mary Pat Moeller and their team of researchers, which elucidates this complicated relationship). I don't want to belabor the point too much (I already have), but I strongly recommend that the authors remove their analysis with aided hearing levels due to concerns about the validity of these data. The data could be analyzed using unaided hearing levels (not ideal, but at least a valid assessment of hearing) or the analysis could be removed altogether from the paper, and the authors could mention in the limitations sections that they were unable to address the role of hearing levels in the analysis.
Author Response

(The authors gave the same response as above.)

Reviewer 3 Report
Comments and Suggestions for Authors
This is an important topic, and it will be useful for the field to have updated data on deaf children’s reading and writing skills. I think that this paper could be highly cited.
My main comments are on the length of the paper (excessive wordiness), the reporting of the results (several of the tables need to be made into figures), and the statistical approach (concern about the number of statistical tests they are running with no correction for multiple comparisons).
Introduction
“This carries significant implications for educational research, policy, and practice, especially as inequities in academic attainment persist (Garberoglio et al., 2021) despite advancements in crucial supports like newborn hearing screening and identification (Yoshinaga-Itano et al., 1998) and early intervention (Meinzen-Derr et al., 2020).” – This sentence is long and important. I strongly recommend breaking it up more so that the reader absorbs it.
Second paragraph is dense and contains several different ideas. I would recommend splitting it into 2 paragraphs. Same comment applies to the fourth paragraph.
It’s important, however, to distinguish between quality access to technology and quality access to actual language, as the mere use of a device does not guarantee optimal language input (Carrigan & Coppola, 2020), nor should early access to technology be considered the dominant variable influencing language growth (Duchesne & Marschark, 2019). – Loved how the authors phrased this.
I recommend that the authors use subheadings throughout the Introduction to better orient the reader.
The third paragraph on page 3 should be broken up.
Overall, the Introduction is lengthier than it needs to be. In some papers, this is because the authors include irrelevant information. Here, the content of the Introduction is great and relevant, but the individual sentences and paragraphs are a little too wordy.
Methods
The authors have data from a large sample of deaf children
How are Gender and Race/Ethnicity categories sorted in Table 1? It doesn’t seem to be alphabetical or largest–smallest.
Age information is not provided. Is this available?
Education, communication, and audiological information should be presented in a table rather than in text. The current format makes it difficult for readers to find that information quickly.
I recommend editing the Table 2 capture to specify where the ratings are from. Ex: “Table 2. Teacher Ratings of Students’ Proficiency in ASL and Spoken English on a Scale of 1-5 (n=368). 323”
“Each WJ-IV subtest provides both a standard score and a W score. The standard score is norm-referenced, with a mean of 100 and a standard deviation of 15, and it cannot fall below a value of 40 due to the scoring structure of the assessment. In contrast, the W score represents a Rasch-derived equal-interval scale that does not have a truncated range, providing a more sensitive measure of student performance.” – this paragraph should specify which measure the authors will be using for analysis.
The information in Table 4 could be visualized more quickly and in richer granularity using a histogram. The authors can then overlay the histogram with background colors to indicate which scores are 1SD from the mean, etc. (https://stackoverflow.com/questions/9968975/make-the-background-of-a-graph-different-colours-in-different-regions/9969983) or even by changing the color of the bars themselves (https://stackoverflow.com/questions/7027448/change-histogram-bar-colours-greater-than-a-certain-value)
A Sankey chart could help visualize the school placement data
Results
Overall, the results are much wordier than needed. I’ve included several suggestions aimed at this point.
My other major concern with the Results: there are many, many statistical tests being run, and there has been no correction for multiple comparisons. At the very minimum, there needs to be family-wise correction.
“This result suggests a strong linear association between students’ literacy skills. A scatterplot was used to visualize the relationship between students' reading and writing scores (Figure 1), with reading scores on the x-axis and writing scores on the y-axis. The line of equality (y = x) was included to indicate where scores in both would be equal. Notably, the majority of data points fell above this line, suggesting that most students scored higher in writing than in reading.” – This all belongs in the figure caption (not in the main text) and could be shortened.
“Furthermore, the scatterplot pattern indicates an upward trend and high association of reading and writing scores, as indicated by the Pearson correlation coefficient.” This sentence is redundant with a sentence earlier in the paragraph. I recommend removing this one entirely
Figure 1 caption should note that the axes have been truncated.
Figure 1: this graph is pixelated / grainy.
Research Question 2: results should be presented in figures (consider box plots with raw data points overlaid?) rather than tables. The tables are unwieldy for readers.
Figure 2 is problematic. The graph does not show confidence intervals of any sort, the raw data, or even the range of values. The x-axis is a categorical scale, but by showing a line graph (instead of violin plots or box plots), it looks as though that scale is actually continuous, which it isn’t. Finally, the x-axis label should specify (in any language). I would ideally like to see a three-panel version of this with English, ASL, and then top score in any language.
“These findings suggest that previously observed group differences were largely accounted for by variation in language proficiency.” – an absence of a significant effect once controlling for language proficiency in this sample does not mean that there are no group differences. If they authors wish to make this claim, they need to conduct a power analysis or other additional analyses to show that this isn’t merely underpowered to detect group differences. (For what it’s worth, I agree with them, but the existing statistics don’t back this up)
Discussion
Third paragraph of the Discussion makes me want to see some analysis that incorporates age of exposure to ASL / English. Do you still see worryingly low reading scores in deaf signing families – where we suspect that there is minimal language deprivation?
Paragraph beginning at line 635 is very good
“Data from this study indicate that, in third grade, average reading scores are one standard deviation below the mean and continue to decline across subsequent grade levels” – This sentence is misleading. If I’m understanding the data correctly, the raw scores do not decline but the standard scores do – indicating a widening gap. The current sentence sounds as though deaf children’s reading skills decline, when really it seems like they’re just not progressing as quickly.
The paragraph beginning at 687 and the paragraph at 718 are redundant with each other, and I recommend consolidating them into one paragraph in the Limitations section.
“Data Availability Statement: The raw data supporting the conclusions of this article will be made available by the authors on request.” – These types of data availability statements rarely deliver as promised (e.g., Gabelica, Bojčić, Puljak, 2022). It is often easiest to set up data sharing now, while the authors are most familiar with the data, then in 3–5 years when things have been forgotten or researchers have moved on. I strongly strongly suggest that the researchers de-identify their data now, figure out what can/can’t be shared without risking confidentiality, document all of the column names / etc. in any spreadsheets, and upload their data to a permanent repository. Research data platforms like OSF or Databrary can be configured to stay Private or allow selective access.
Author Response

(The authors gave the same response as above.)

Round 2
Reviewer 2 Report
Comments and Suggestions for Authors
I appreciate the effort the authors have made in addressing my suggestions for revisions although I am still unclear why the authors would exclude references from Werfel and Lund's ELLA studies that relate to literacy. They also didn't include any citations from the LOCHI study, which is another important large-scale prospective longitudinal study on outcomes of children with hearing loss (I didn't mention this in my prior review, which was my mistake). In general, the references are adequate and there isn't a need for additional citations.
My larger concern regarding the measures of hearing levels or auditory access still stands. Most importantly, it's still unclear to me how the authors are calculating hearing levels, based on the methods section. Are they calculating a three or four frequency pure tone average for the better ear and using that measure in their analysis to group the participants into categories of mild, moderate, etc? Is better ear aided or unaided PTA used as a continuous independent variable? I'm pretty confident this is the case but it's not clear in the manuscript. How did they obtain the data on hearing levels and who collected these data? There is also no information about hearing levels in the demographic table so it's unclear what the mean aided and unaided thresholds were. More information is necessary in the methods section to describe the way in which they measured hearing levels.
I still feel that by measuring aided thresholds with a hearing aid or a CI or unaided thresholds, the authors are comparing apples to oranges to grapes. The authors would be better off just using unaided thresholds for all of the participants than trying to combine these different measures of hearing sensitivity together. The authors stated that they were combining these different measures so they could represent "the level of hearing sensitivity accessible to the child in real-world conditions", but the stimuli that they are using to measure this is not representative of the real world (i.e., the thresholds would be presumably measured with pure tones for unaided thresholds and warble tones in sound field for aided thresholds - neither of which is what a child is going to encounter in the real world). I appreciate that the authors did add a caveat to the limitations section regarding the issues with aided audiograms which at least partially addresses my concerns. I won't ask for any additional revisions here but I hope that in future studies, the authors are more cognizant of the need to accurately measure auditory access using valid techniques - I recognize that this is a very difficult task when you are combining children with different hearing technologies, but if the conclusion is going to be that children who are deaf/hard of hearing can't acquire language via hearing technology because it doesn't provide quality language access, the measures for doing so have to be better than functional hearing measures, which aren't ecologically valid.
Author Response
See attached document.

Reviewer 3 Report
Comments and Suggestions for Authors
The authors have largely addressed my concerns in the Introduction and Discussion sections, but I continue to have concerns over the reporting of the results.
The authors seem to disagree with my suggestion to move some of the tables into figures because, they argue, the figures would be less clear. Looking at Figure 2, I think I better understand our disconnect; their previous table was clearer than this set of boxplots.
If the authors wish to keep figure 2 as a figure, here are my recommendations:
- Facet the graph by reading/writing, unless the authors want the main comparison to be between those two scores.
- Overlay the raw data points on the graph (rather than just the outliers – which are labelled with participant number, it seems??). Use semi-transparent dots. This will show readers the range and distribution.
- I’ve included an example graph (sometimes called a pirateplot) that shows that range, distribution, and central tendencies at a glance

I continue to believe strongly that swapping out the tables for figures (if the authors take the time to design easy-to-interpret, informative graphs) will make this paper much more digestible to the reader who wants to know at a glance, how much does mild vs. moderate deafness affect reading scores?
It seems like the authors are writing for an education audience rather than a behavioral sciences audience, and believe that the exact numbers are more important to readers than the patterns between the numbers. If that is what the authors would like to provide, these data can be provided as tables at the end of the article or in supplemental materials.
Data Availability Statement
The authors should say this then! Because if this is the plan, and there is currently an embargo, then the current statement is inaccurate. I would recommend even creating the (empty) public repository now, so that they can provide the link where the data will be. I would likely say something like: “The data collected for this study are from a large federally-funded study. There is a temporary embargo on the data, but the data will be publicly available at this link (your repo link) following final analysis of the data (anticipated availability: late 2026)”.
Author Response
See attached document

Round 3
Reviewer 3 Report
Comments and Suggestions for Authors
The authors have adequately addressed my concerns.
Author Response
.